# Study on the Impact of Delayed Retirement on the Sustainability of the Basic Pension Insurance Fund for Urban Employees in China

**Guiling Zhao** [1,2], **Deyu Zhou** [3] **and Yunpeng Fu** [1,*]

1    School of Economics, Liaoning University, Shenyang 110036, China; zhaoguiling@hbfu.edu.cn
2    School of Finance and Investment, Hebei Finance University, Baoding 071051, China
3    School of Mechanical Engineering, University of Science and Technology Beijing, Beijing 100083, China; zxzhoudeyu1999@126.com
*    Correspondence: fyp@lnu.edu.cn; Tel.: +86-13066538561

**Abstract:** With the aging of China's population, the problem of pension security has become more and more prominent, and whether delayed retirement can effectively alleviate the pension fund gap and ensure the sustainability of the pension fund has become the focus of social concern. This study predicts the income and expenditure of urban workers' basic pension insurance fund from 2021 to 2050 by constructing an actuarial model of pension insurance fund income and expenditure, and simulates the effect of delayed retirement policy. The prediction results show that under the existing system, the basic pension insurance fund for urban workers will have a shortfall for the first time in 2027, and the shortfall will expand year by year. Compared with the non-implementation of delayed retirement policy, the simulation of the implementation of a delayed retirement program delayed the emergence of the fund gap until 2029, and the forecast period of the pension fund gap significantly narrowed, indicating that delayed retirement policy has a certain positive impact on alleviating the pressure of pension payments, but delayed retirement cannot completely eliminate the pension fund gap. In view of this, this paper suggests that a progressive and flexible delayed retirement policy should be introduced as soon as possible to better adapt to the needs of different groups. At the same time, differentiated policies should be formulated for different groups of people and a pension incentive mechanism for delayed retirement should be set up to improve public acceptance of delayed retirement policy. In addition, delayed retirement policy should be combined with other measures, such as lowering the corporate contribution rate and enhancing the value-added capacity of the pension fund, so as to ensure the sustainability of the pension fund.

**Keywords:** population aging; pension fund gap; delayed retirement; sustainability

## 1. Introduction

As China's aging population continues to increase, the degree of population aging is becoming increasingly severe [1]. At the same time, the average life expectancy of the population is increasing year by year, and urban workers, as the main expenditure subject of basic pension insurance, make the payment pressure of the pension fund more and more intense [2,3], thus affecting the sustainability of the fund [4]. In recent years, the pension insurance fund for China's urban workers has gradually shown an income and expenditure gap [5], and the growth rate of accumulated balance also demonstrated a trend of sharp decline [6]. In this case, the study of the impact of delayed retirement on the sustainability of the pension fund is of urgent and very important practical significance. The notice on the Development of National Elderly Care and Old-age Support Service System for the 14th Five-Year Plan, issued by the State Council in February 2022, explicitly states that a gradual delay in the legal retirement age should be implemented to alleviate the pressure on pension payments. The report of the 20th Party Congress also proposed to improve the

national coordination system of basic pension insurance and implement a gradual delay of the statutory retirement age.

Many scholars have long studied and explored the issue of delayed retirement, and most scholars believe that delayed retirement can effectively reduce the pension gap and improve the sustainability of the pension fund [7–9]. They believe that with the aging of the population, lengthening the working life of workers can increase pension income [10], reduce pension expenditures [11], and narrow the pension fund gap [12], thus easing the pressure faced by the pension system [13–16]. Börsch-Supan and Berkel (2004) argue that the increasing aging of the population and the precarious financial situation of the public pension system will lead to a serious financial crisis of pension funds. In order to avoid a crisis, the extension of the retirement age must be used as an adjustment factor in pension reform [7]. Some scholars have further investigated the internal and external motivations and motivational hierarchy of the elderly to continue working after being eligible for retirement [17]. In addition, there are significant differences in extending time, extending retirement, and among different socioeconomic groups, industries, and occupations [18]. Based on the extended time perspective, Tian and Zhao (2016) found that if the statutory retirement age is delayed by five years, the emergence of the basic pension fiscal gap may be delayed by about 20 years, and the median basic pension deficit will be reduced by about 64.25% in 2087 [6]. Based on the empirical data of the United States, Biggs (2010) used a demographic microsimulation model to estimate that a retirement age extension from 62 to 65 would extend the life of the trust fund by five years; increase social security benefits by 16 percent and private pensions by USD 7500 per year when individuals retire; and raise GDP by about 5 percent, adding billions of dollars to the economy and tax revenues [19]. Based on the retirement approach perspective, Hu et al. (2023) argued that a rolling retirement age adjustment policy is more appropriate for a healthy economy than a one-step retirement age adjustment program [20].

Another group of scholars hold a different view. Chen et al. (2020) argued that delayed retirement may not achieve the desired effect; the implementation of policies is limited and treats the symptoms rather than the root cause, and cannot be the main method to bridge the gap [21]. Precarious working conditions, family care responsibilities, poor health, and age discrimination make it difficult or impossible for many people to work into their 60s or beyond, making it more difficult to implement delayed retirement policies [22,23]. In addition, retirees with lower levels of education are more likely to stop working for medical reasons [24]. Some scholars also believe that delayed retirement policies may also lead to increased competition in the labor market, making it more difficult for young people to find employment and also increasing the employment pressure on the elderly, which may have a negative impact on economic development and social stability, and thus affect the sustainability of the pension system [25]. However, Munnell and Wu (2012), using U.S. data, found that increased employment of older people does not reduce employment opportunities or wages for younger people. On the contrary, the employment of older people may also help to increase the employment rate of younger people [26]. Burtless (2013) also used U.S. data to demonstrate that workers between the ages of 60 and 74 are more productive than the younger average worker, and that workers remain in the labor force for longer periods of time the more productive they are [27]. Oyaro et al. (2015) found through an empirical study of the transport and logistics industry in Australia that the supply of older workers is not the main barrier to young people gaining employment, but rather the recruitment practices and strategies used by employers to avoid the cost of training younger workers [28].

Overall, the impact of delayed retirement policy on the sustainability of pension funds is a complex issue that requires a comprehensive consideration of various factors. The research results of scholars provide different perspectives and analyses, which provide a reference and decision-making basis for policymakers. But most of the existing results study the impact of delayed retirement on the coordinated account of the pension fund. In fact, delayed retirement also has an impact on personal accounts, the study of which is not

comprehensive enough. In addition, when conducting the modeling of pension income and expenditure, existing studies consider the factor of government financial subsidy less, which leads to biased research results.

Based on this, this paper contributes in the following three aspects: Firstly, financial subsidies are included in the actuarial model of the income and expenditure of the urban workers' pension insurance fund. The prediction results are closer to the reality. Secondly, it expands the research scope. When this paper studies the impact of delayed retirement on the sustainability of the pension fund, the sources of the pension fund include not only the coordinated account, but also individual accounts, and take into account the factor of the government's financial subsidies, which makes the study more comprehensive. Thirdly, we construct an actuarial model of the income and expenditure of the urban workers' pension insurance fund, including the coordinated account, individual accounts, and government subsidies, to predict and compare the income, expenditure, and surplus of pension insurance under the two policies of delayed retirement and non-delayed retirement, to explore the impact of delayed retirement on the sustainability of the urban workers' pension insurance fund, and to provide a scientific basis for the government to formulate a more accurate retirement policy to ensure the sustainability and stability of the pension fund.

## 2. Theoretical Analysis

Delaying the retirement age is a policy measure that many countries around the world are considering or have already implemented, aimed at addressing population aging, extending the duration of labor market participation, alleviating the financial pressure on the pension system, and improving the sustainability of pension funds. Theoretically, delaying retirement affects the sustainability of pension funds through adjustments to the pension system, changes in the structure of the labor market, a reduction in dependency on fiscal subsidies, and increases in fiscal revenue.

*2.1. Delayed Retirement Policy Affects the Sustainability of Pension Funds through Adjustments to the Pension System*

Firstly, the implementation of delayed retirement policy leads to an increase in the number of contributors and the amount of contributions. Delayed retirement means that employees will work for an extended period, thereby increasing the number of contributors and the amount of contributions. In the pension system, employee contributions are the main source of fund income. With the extension of contribution years, the amount of funds accumulated in individual accounts will increase, thereby improving the accumulation scale and payment capacity of the pension insurance fund, ensuring the sustainability of the pension system [29]. Secondly, the delayed retirement policy will extend the number of years in which employees contribute. If the number of years for receiving benefits remains unchanged or grows incrementally, the total number of years that employees receive pensions will decrease. Therefore, the total amount of pensions that the pension insurance fund needs to pay will be reduced, thereby reducing the payment pressure on the fund. Moreover, reducing the number of years for receiving benefits helps to increase the accumulation scale of the pension insurance fund, further enhancing the fund's payment capacity [14]. Thirdly, the delayed retirement policy will increase the accumulated amount in individual accounts, improving the pension replacement rate. The increase in the pension replacement rate helps to ensure a basic standard of living for retirees and improves their quality of life [30]. Fourthly, the delayed retirement policy will increase the investment time of the pension fund, helping to improve investment returns. Investment returns are an important source of income for pension funds, and the increase in investment returns will strengthen the payment capacity of the pension fund and improve its sustainability.

*2.2. Delayed Retirement Policy Affects the Sustainability of the Pension Fund by Changing the Labor Market Structure*

The implementation of the policy helps to optimize the labor market structure. On one hand, the policy increases the supply of elderly labor in the market, which helps to

alleviate the supply and demand contradictions in China's labor market; on the other hand, it contributes to the improvement of employees' vocational skills and experience accumulation, enhancing the overall quality of the labor market. Optimizing the labor market structure helps to increase the speed of economic growth, add to social wealth, and provide strong support for the sustainability of the pension system [31]. However, the employment of older workers may also exert some pressure on the job prospects of younger workers [25]. However, this impact can be mitigated if labor market policies can be rationally adjusted, for example, by improving the quality of education and vocational training and promoting the structural optimization of the workforce. If labor market policies can be adjusted reasonably, for instance, by improving the quality of education and vocational training and promoting the optimization of the labor force structure, such an impact can be mitigated. In addition, the delayed retirement policy also helps to increase the employment rate of women and promote gender equality. The improvement of the labor market will bring more sources of contributions to the endowment insurance fund, further enhancing the sustainability of the fund.

*2.3. Delayed Retirement Policy Strengthens the Sustainability of the Pension Fund by Reducing Dependence on Fiscal Subsidies and Increasing Fiscal Revenue*

The implementation of the delayed retirement policy has significantly reduced the dependency of the pension fund on government fiscal subsidies, which is beneficial for the long-term sustainable development of the pension fund. By extending the working years and contribution period of employees, the policy increases the number and base of pension contributors, thereby raising the revenue of the pension insurance fund. This measure not only enhances the financial autonomy of the fund but also helps to reduce the dependency on government finances, ensuring the stable operation and sustainable development of the pension fund [16]. Moreover, the delayed retirement policy helps to increase labor participation rates, making the labor market more active. The activity of the labor market is conducive to promoting economic development, raising the employment rate, and further increasing tax revenue. The increase in tax revenue can provide the government with more financial resources to support the development of the pension fund, thereby enhancing the financial soundness and sustainability of the pension fund.

Based on the above theoretical analysis, we propose the following research hypotheses:

**Hypothesis 1:** *Delaying retirement can increase the income level of the pension fund.*

**Hypothesis 2:** *Delaying retirement can reduce the expenditure level of the pension fund.*

**Hypothesis 3:** *Delaying retirement is beneficial for enhancing the sustainability of the pension fund.*

### 3. Construction of Theoretical Model

China's current basic pension insurance system for urban workers is based on a combination of a unified account, i.e., a combination of a social coordinating account and an individual account. The basic pension scheme in our country is a mixed pension scheme because the social pooling basic pension belongs to the DB type, while the individual account pension belongs to the DC type. In reality, the income of the basic pension insurance fund for urban workers in China mainly consists of three parts: the income from the social integrated account, the income from the individual account, and the financial subsidies from the government. Therefore, in this paper, the income of the basic pension insurance fund for urban workers is measured by the three parts of the integrated account, individual account, and financial subsidies. For the sake of simplicity of calculation, the cases of early retirement and mid-term withdrawal are not taken into account.

*3.1. Pension Fund Income Modeling*

3.1.1. Model of Income from Pension Insurance Benefits in Social Integration Accounts

The growth of the average annual wage income of urban workers is influenced by social and economic growth and the improvement of the comprehensive ability of individuals, so the urban workers' pension contribution wages will also be affected. This paper assumes that the average wage level of employees due to the social and economic growth has a growth rate of $\chi$, so the average wage level of urban in-service workers at the age of $i$ in the year of $t$ can be represented as

$$W_i(t) = W_{i-1}(t-1) \times (1+\chi) \tag{1}$$

Among them,

$$W_{i-1}(t-1) = W_{i-2}(t-2) \times (1+\chi) \tag{2}$$

And so the following result can be obtained:

$$W_i(t) = W_b(t-i+b) \times (1+\chi)^{i-b} \tag{3}$$

where $W_i(t)$ denotes the average wage level of urban active workers aged $i$ in year $t$; $W_{i-1}(t-1)$ denotes the average wage level of urban active workers aged $i-1$ in year $t-1$; $W_b(t-i+b)$ denotes the average wage level of urban active workers aged $b$ in year $t-i+b$, i.e., the average wage level of urban active workers aged $i$ in the first year of their employment in year $t$; and $b$ denotes the initial age of the worker's enrollment in the program.

The number of contributors to the pension fund of urban active workers at the age of $i$ in year $t$ can be expressed as the average of the number of contributors at the beginning of year $t$ (i.e., at the end of the year $t-1$) and the number of contributors at the end of year $t$. It can be expressed by the following formula:

$$Ne_i(t) = \frac{I_{i,t} + I_{i-1,t-1}}{2} \tag{4}$$

where $Ne_i(t)$ represents the number of contributors to the pension fund of urban active workers at the age of $i$ in year $t$; $I_{i,t}$ represents the number of contributors at the end of year $t$; and $I_{i-1,t-1}$ represents the number of contributors at the beginning of year $t$.

Then, starting from the base period, the income of the pension insurance premium in the social coordinated account in year $t$ is equal to the number of contributors to the pension insurance premiums of urban active workers at different ages in that year multiplied by the corresponding average wage level multiplied by the pension insurance premium contribution rate of the coordinated account multiplied by the pension insurance premium compliance rate of the coordinated account. The model can be represented as follows:

$$R(t) = \sum_{i=b}^{M-1} Ne_i(t) \times W_i(t) \times Cr_2 \times O_t \tag{5}$$

where $R(t)$ is the income of pension insurance premiums of the social coordination account in year $t$; $b$ represents the initial age of participation of the employees; $M$ represents the retirement age of the employees; $Ne_i(t)$ represents the number of contributors to the pension insurance premiums of the urban active employees at the age of $i$ in year $t$; $W_i(t)$ represents the average level of wages of the urban active employees at the age of $i$ in year $t$; $Cr_2$ represents the rate of pension insurance premium contributions to the coordination account; and $O_t$ represents the rate of pension insurance premium compliance of the coordination account in year $t$.

By substituting Formulas (3) and (4) into Formula (5) and arranging the terms, we obtain

$$R(t) = \sum_{i=b}^{M-1} \frac{I_{i,t} + I_{i-1,t-1}}{2} \times W_b(t-i+b) \times (1+\chi)^{i-b} \times Cr_2 \times O_t \tag{6}$$

And due to the differences in the statutory retirement age for men and women currently set in China, this model can be further refined as

$$\begin{aligned}
R(t) = &\sum_{i=b}^{M_m-1} \left[ \frac{I_{m,i,t}+I_{m,i-1,t-1}}{2} \times W_b(t-i+b) \times (1+\chi)^{i-b} \times Cr_2 \times O_t \right] \\
&+ \sum_{i=b}^{M_f-1} \left[ \frac{I_{f,i,t}+I_{f,i-1,t-1}}{2} \times W_b(t-i+b) \times (1+\chi)^{i-b} \times Cr_2 \times O_t \right]
\end{aligned} \tag{7}$$

In this case, variables with subscript $m$ indicate males and those with subscript $f$ indicate females.

### 3.1.2. Individual Account Pension Income Model

From the base period, the income of the individual account pension insurance premiums in year $t$ is the product of the number of urban active workers of different ages who contributed to the pension insurance premiums in that year, the corresponding average salary level, the growth rate of the premiums in the individual account since participation in the program, the rate of contribution to the pension insurance premiums in the individual account, and the rate of compliance with the contribution to the pension insurance premiums in the individual account. The model can be represented as

$$R(t)' = \sum_{i=b}^{M-1} Ne_i(t) \times W_i(t) \times (1+r)^{M-i} \times Cr_1 \times V_t \tag{8}$$

where $R(t)'$ is the income of individual account pension insurance premiums in year $t$; $b$ represents the initial age of employees' enrollment; $M$ represents the retirement age of employees; $Ne_i(t)$ represents the number of contributors to the pension insurance premiums of urban active employees at the age of $i$ in year $t$; $W_i(t)$ represents the average level of wages of urban active employees at the age of $i$ in year $t$; $r$ represents the rate of return on the fund; $Cr_1$ represents the rate of contribution to pension insurance premiums in individual accounts; and $V_t$ represents the rate of compliance with the pension insurance premiums in individual accounts in year $t$.

By substituting Formulas (3) and (4) into Formula (8) and arranging the terms, we obtain

$$R(t)' = \sum_{i=b}^{M-1} \frac{I_{i,t} + I_{i-1,t-1}}{2} \times W_b(t-i+b) \times (1+\chi)^{i-b} \times (1+r)^{M-i} \times Cr_1 \times V_t \tag{9}$$

And due to the differences in the statutory retirement age for men and women currently set in China, this model can be further refined as

$$\begin{aligned}
R(t)' = &\sum_{i=b}^{M_m-1} \left[ \frac{I_{m,i,t}+I_{m,i-1,t-1}}{2} \times W_b(t-i+b) \times (1+\chi)^{i-b} \times (1+r)^{M-i} \times Cr_1 \times V_t \right] \\
&+ \sum_{i=b}^{M_f-1} \left[ \frac{I_{f,i,t}+I_{f,i-1,t-1}}{2} \times W_b(t-i+b) \times (1+\chi)^{i-b} \times (1+r)^{M-i} \times Cr_1 \times V_t \right]
\end{aligned} \tag{10}$$

In this case, variables with subscript $m$ indicate males and those with subscript $f$ indicate females.

### 3.1.3. Financial Subsidies

Considering the government's financial subsidies as one of the important sources of urban workers' pensions, the proportion of financial subsidies in the annual income of the basic pension insurance fund for urban workers is relatively stable, so the proportion of financial subsidies in the income of the basic pension insurance fund for urban workers can be used to make a prediction of the future government's financial subsidies. Take $G_{(t)}$ as the government's financial subsidy to the basic pension insurance fund for urban workers in year $t$, and $n$ as the proportion of the financial subsidy to the total income of the pension insurance fund for urban workers. Then, the proportion of the government's financial subsidy to the basic pension fund for urban workers in year $t$ will be

$$n = \frac{G(t)}{R(t) + R(t)' + G(t)} \tag{11}$$

The above equation gives us the government's financial subsidy to the basic pension fund for urban workers in year $t$:

$$G(t) = \frac{n[R(t) + R(t)']}{1 - n} \tag{12}$$

Then, the total income of the basic pension insurance fund for urban workers is equal to the income from the social coordination account plus the income from the individual account plus the government's financial subsidy, so the income of the pension insurance fund for urban workers is

$$F(t) = R(t) + R(t)' + G(t) = \frac{R(t) + R(t)'}{1 - n} \tag{13}$$

### 3.2. Model of Social Pension Expenditure

The number of retired people receiving pensions at the age of $i$ in year $t$ can be expressed as the average of the number of people at the beginning of year $t$ (i.e., at the end of year $t - 1$) and the number of people at the end of year $t$. It can be expressed by the following formula:

$$Nr_i(t) = \frac{U_{i,t} + U_{i-1,t-1}}{2} \tag{14}$$

where $Nr_i(t)$ represents the number of pensioners who retired at age $i$ in year $t$, $U_{i,t}$ represents the number at the end of year $t$, and $U_{i-1,t-1}$ represents the number at the beginning of year $t$.

Expenditures for social pension benefits were

$$E(t) = \sum_{i=M}^{w-1} Nr_i(t) \times P_{t,i} \tag{15}$$

where $E(t)$ denotes social pension expenditures; $w$ denotes the average age of death of workers; and $P_{i,t}$ denotes the pension of workers who retired at age $i$ in year $t$.

Adjust the pension level according to the state of economic development, where $\alpha$ represents the annual adjustment rate of the pension; then,

$$P_{i,t} = P_{M,t-i+M}(1 + \alpha)^{i-M} \tag{16}$$

where $P_{M,t-i+M}$ denotes the average pension received by a retired worker aged $M$ in year $t - i + M$, i.e., the average pension received by a retired worker aged $i$ in year $t$ at the beginning of the first year of their retirement.

Also, because the pension replacement rate is the ratio of the pension of a participant in the first year of retirement to the level of the participant's average wage just before the retirement year, denoted by *R*, then

$$R = \frac{p_{M,t-i+M}}{W_{M-1}} \tag{17}$$

where $W_{M-1}$ denotes the average salary of retired urban workers in the year before their retirement.

Equations (14), (16) and (17) are obtained by bringing in (15) and organizing

$$E(t) = \sum_{i=M}^{w-1} \frac{U_{i,t} + U_{i-1,t-1}}{2} \times R \times W_{M-1} \times (1+\alpha)^{i-M} \tag{18}$$

This model can be further refined due to the different statutory retirement ages for male and female workers and the average life expectancy for men and women currently set in China:

$$
\begin{aligned}
E(t) = {} & \sum_{i=M_m}^{w_m-1} \left[ \frac{U_{m,i,t} + U_{m,i-1,t-1}}{2} \times R \times W_{M-1} \times (1+\alpha)^{i-M} \right] \\
& + \sum_{i=M_f}^{w_f-1} \left[ \frac{U_{f,i,t} + U_{f,i-1,t-1}}{2} \times R \times W_{M-1} \times (1+\alpha)^{i-M} \right]
\end{aligned}
\tag{19}
$$

*3.3. Urban Workers' Basic Pension Insurance Pension Revenue and Expenditure Gap Measurement Model*

3.3.1. Current Urban Workers' Basic Pension Insurance Pension Income and Expenditure Gap

The current fiscal gap of the basic old-age insurance for urban workers equals the income minus the expenditure of the basic old-age insurance for urban workers. The income of the basic old-age insurance for urban workers in the current period equals the income from the social pooling fund plus the income from the individual account fund plus the fiscal subsidies from the government. The calculation formula is

$$Q(t) = F(t) - E(t) = \frac{R(t) + R(t)'}{1-n} - E(t) \tag{20}$$

where $Q(t)$ indicates the difference between the income and expenditure of the basic pension fund for urban workers in the current period. A positive value of $Q(t)$ indicates that the current pension fund income is greater than the expenditure, and the pension fund is sufficient. A value of $Q(t)$ indicates that the current pension fund income equals the expenditure, and the pension fund is in a balanced state. A negative value of $Q(t)$ indicates that the current pension fund income is less than the expenditure, and the pension fund is insufficient, affecting the sustainability of the pension fund.

3.3.2. Accumulated Balance of Urban Workers' Basic Pension Insurance Pensions

The accumulated balance of the current urban employees' basic pension insurance fund is equal to the income and expenditure of the current fund plus the accumulated balance of the fund of the previous period, which is expressed by the following formula:

$$A(t) = Q(t) + A(t-1) \tag{21}$$

where $A(t)$ denotes the accumulated balance of the basic pension fund for urban workers in year *t*. A positive value of $A(t)$ indicates that the accumulated income of the pension fund is greater than the accumulated expenditure in period t, and the accumulated pension fund is sufficient. A value of $A(t)$ indicates that the accumulated income of the pension fund equals the accumulated expenditure in period t, and the accumulated pension fund

is in a balanced state. A negative value of $A(t)$ indicates that the accumulated income of the pension fund is less than the accumulated expenditure in period t, and there is a gap in the accumulated pension fund, which exists a problem with the sustainability of the pension fund.

Based on historical data, we assign specific values to the relevant parameters for the years 2021 to 2050. Subsequently, by applying Formula (20), we predict how the balance between the revenue and expenditure of China's urban workers' basic old-age insurance fund will change before and after the implementation of the delayed retirement policy. Moreover, using Formula (21), we will also predict the development of the fund's cumulative surplus under this policy. Through these forecasts, we can deeply explore the potential impact of delayed retirement on the sustainability of China's urban workers' basic old-age insurance fund, providing decision-making support for policymakers and ensuring the long-term stable operation of the pension insurance system.

## 4. Parameter Setting and Retirement Plan Design

### 4.1. Parameter Setting

#### 4.1.1. Total Population and Population by Sex

Based on the data from the *Seventh National Population Census* in 2020, using the PADIS-INT V1.2.2.5 version of the population projection software, the start and end years are set as 2020 and 2050, respectively, with a parameter adjustment interval of 1 year. The United Nations model life table is used. Setting relevant parameters is an essential step in the process.

Step 1: Set the number of the starting population.

The *Seventh National Population Census* in 2020 is the closest one to the present, which happens every ten years. The data are new and can reflect the reality more accurately, so the parameter of the population starting number is selected as the base period data of the age-specific and sex-specific population number in the Seventh National Population Census in 2020.

Step 2: Set the fertility patterns.

Generally, the age-specific fertility rate is relatively stable and will not undergo large fluctuations. The fertility pattern selects the fertility rate of women of childbearing age in each age group from 15 to 49 years in 2020.

Step 3: Set the fertility level.

The *National Population Development Plan (2016–2030)* mentions that China's total fertility rate is expected to reach about 1.8 by 2030, and according to the relevant surveys by the National Bureau of Statistics, the rate of women of childbearing age in China who wish to have children is also 1.8. Therefore, this article assumes that China may introduce policies to encourage childbirth, stimulating the fertility potential of women of childbearing age, so that the total fertility rate will rise to 1.8 by 2030 (the level of willingness to bear children), and then remain at 1.8 by 2050.

Step 4: Set the birth sex ratio.

Under normal circumstances, the sex ratio at birth is determined by biological laws, with a normal range of 102–107. China's sex ratio at birth has been on the high side for a long period of time, and the government and related departments have taken many positive measures to intervene, and in recent years, the rising trend of China's sex ratio at birth has slowed down or even begun to gradually decline, and will return to normal levels by about 2050 [32]. In this paper, the sex ratio at birth is based on the viewpoint of Chao et al. (2021) [29] that China's sex ratio at birth will reach 106 in 2050, and the sex ratio at birth in the intermediate years is smoothed.

Step 5: Set the migration pattern and the migration level.

Since the level of in-migration and out-migration in China is not large, the prediction of future population can be regarded as operating in a closed environment, so population migration is not considered, and the relevant parameters for each year are set to 0 [33,34].

Step 6: Set the life expectancy.

According to the experience of developed countries, it is known that the life expectancy of the population will continue to increase with the improvement of the standard of living. In this paper, the population life expectancy is based on the projection of China's population life expectancy in 2050 in *World Population Prospects 2019*, which is 78.8 for males and 82.9 for females, with smoothing of the data in the intermediate years.

Based on the parameters set above, the total population by age and by sex, as well as the annual total population and population size by sex, are projected for the next 30 years in China.

### 4.1.2. Working Age and Retirement Age

The minimum age of employment stipulated in China's labor law is 16 years old, but according to the data from the *2020 China Population and Employment Statistical Yearbook*, the proportion of China's employed persons aged 16–19 years old to all employed persons is only 1%, so this paper assumes that an urban worker participates in the workforce at the age of 20, and after enrolling in pension insurance, they continuously pay the contributions until retirement with no breaks in the middle of the period. The retirement age for men is 60 years old. The retirement age for female workers is 50 years old. The retirement age for female cadres including civil servants, career organizations, etc., is 55 years old. Considering the retirement regulations for female cadres, this paper assumes that the proportion of female cadres among women aged 50–54 is $\gamma$, and that they retire at the age of 55. There will be a total of 7.1 million civil servants and 31 million career employees, both totaling 38.1 million people, accounting for about 8.29% of the 459.31 million people employed in cities and towns nationwide in 2022. Assuming $\gamma = 10\%$ in this paper, the number of employed and retired women per year is shown below.

$$I_{f,t} = \sum_{i=20}^{49} F^1_{f,i,t} + \beta \times \sum_{i=50}^{54} F^2_{f,i,t} \tag{22}$$

$$U_{f,t} = (1 - \beta) \times \sum_{i=50}^{54} F^2_{f,i,t} + \beta \times \sum_{i=55}^{100} F^3_{f,i,t} \tag{23}$$

where $I_{f,t}$ denotes the number of employed women in year $t$, $U_{f,t}$ denotes the number of retired women in year $t$, and $F^1_{f,i,t}$, $F^2_{f,i,t}$, and $F^3_{f,i,t}$ denote the number of employed women aged 20–49 and 50–54, and retired women aged 55–100, respectively. $\beta$ denotes the proportion of female cadres among women aged 50–54.

### 4.1.3. Maximum Age of Survival

With the development of science and technology, the life expectancy of human beings in the future will be slowly extended to a certain extent, but this growth is very slow; when it reaches a certain point, the room for growth will gradually become smaller. Therefore, according to the *China Life Insurance Industry Experience Life Tables (2000–2003)*, this paper assumes that the maximum age of survival for urban worker retirees is 100 years old.

### 4.1.4. Average Annual Wage Growth Rate

The annual average wage growth rate is the growth rate of the average wage level of urban workers as a result of social and economic growth. According to the relevant data from the National Bureau of Statistics, the trend of China's average wage growth rate is the same as the trend of GDP. This paper refers to Liu et al.'s (2022) [35] prediction of China's future GDP, and sets the future average annual growth rate of urban workers at 7.9% in

2021–2025, 7.0% in 2026–2030, 6.4% in 2031–2035, 5.5% in 2036–2040, 4.4% in 2041–2045, and 3.5% in 2046–2050.

### 4.1.5. Population Urbanization Rate

The population urbanization rate is a measure of urbanization, expressed as the proportion of the urban population to the total population. According to the *China Statistical Yearbook* of past years, China's urbanization rate from 2010 to 2020 would be 49.68%, 51.27%, 52.57%, 53.73%, 54.77%, 56.10%, 57.35%, 58.52%, 59.58%, 60.60%, and 63.89%, respectively. From the Northam curve, it can be seen that the urbanization process is divided into three parts. The initial stage for the urbanization rate is less than 30%; the growth of urbanization rate in this stage is slow. In the middle stage, the urbanization rate is at 30% to 70%; the growth of the urbanization rate in this stage is rapid, and China is currently in this stage. In the late stage, the growth of the urbanization rate is very slow, and when the urbanization rate is around 80%, it will be in a state of near convergence. Since the current process of China's urbanization has entered the middle and late stages, the growth rate of urbanization is bound to not maintain the previous trend, so regarding the value of the future urbanization rate, we refer to the prediction of China's urbanization rate by Gu et al. (2017) [36], and set the urbanization rates of 2030 and 2050 to 70% and 80%. The average annual urbanization growth rate in China from 2020 to 2030 is 1.33%, and from 2031 to 2050, it is projected to be 0.67%. Based on this, we can calculate the urbanization rate for the period from 2021 to 2050.

### 4.1.6. Employment Rate of the Urban Population

The employment rate is an indicator of the degree of employment of the labor force, referring to the percentage of employed persons in the sum of employed persons and persons awaiting employment. According to the *Statistical Bulletin on the Development of Human Resources and Social Security* of past years, the urban unemployment rate from 2010 to 2020 can be obtained as 4.10%, 4.10%, 4.10%, 4.05%, 4.09%, 4.05%, 4.02%, 3.90%, 3.80%, 3.62%, and 4.24%, respectively, with an average urban unemployment rate of 4.01%. The unemployment rate is 4.01%. Accordingly, this paper sets the unemployment rate of urban workers in 2021–2050 at 4.00% and keeps it unchanged; then, the employment rate of urban workers in 2021–2050 is 96.00%.

### 4.1.7. Compliance Rate for Basic Pension Insurance for Urban Workers

The compliance rate refers to the proportion of urban workers' basic pension insurance that actually pays contributions to the insured. Many enterprises in China experience the behavior of fee evasion, and the phenomena of non-payment, underpayment, and delinquency of social security fees are very common. The compliance rate of basic pension insurance for urban workers in China in 2020 is 72.03%. This paper combines the assumption of Xie et al. (2020) [5] that the coverage of China's pension insurance system would realize full coverage in 2020, and that social security fees would be collected by the tax department from 2019; the collection rate of urban workers' basic pension insurance fund has increased. So, this paper assumes that the collection rate of urban workers' basic pension insurance fund is 85% from 2021 to 2050.

### 4.1.8. Contribution Rate for Basic Pension Insurance for Urban Workers

In order to adapt to the economic and social development situation, the government has continued to increase its efforts to reduce taxes and fees. Since 1 May 2019, the *Notice of the General Office of the State Council on Issuing a Comprehensive Program to Reduce the Rate of Social Insurance Fees* has shown that the unit's contribution will continue to be reduced to 16%, and that of the individual will remain unchanged at 8%. Therefore, in this article, the contribution rate for the social integration part is set at 16%, and the contribution rate for the individual account is 8%.

### 4.1.9. Pension Replacement Rate

In the *Convention on Minimum Standards of Social Security*, a minimum pension replacement rate of 55 percent is stipulated, with a rate of less than 50 percent implying a significant decline in the standard of living of workers after retirement. According to the *Decision of the State Council on the Establishment of a Unified Basic Pension Insurance System for Enterprise Employees* in 1997, China's current basic pension insurance system has a target pension replacement rate of 58.5%, while the actual level has shown a downward trend in recent years. According to the relevant data of the National Bureau of Statistics, the pension replacement rate of China's basic pension insurance for urban workers from 2010 to 2020 was calculated to be 57.05%, 55.37%, 54.69%, 52.91%, 53.33%, 51.96%, 51.35%, 57.48%, 57.34%, 55.50%, and 53.74%, respectively. The trend of pension replacement rate changes in the past ten years is relatively stable, and the overall trend is low. This article refers to the International Labor Organization's provision on the minimum standard of 55% for the basic pension replacement rate for urban workers, and assumes that the pension replacement rate of China's basic pension insurance fund for urban workers will be 55% from 2021 to 2050.

### 4.1.10. Pension Adjustment Rate

In the long run, the pension adjustment rate is subject to changes to pension benefit adjustment policies, economic growth, and other factors. According to the *Notice of the Ministry of Human Resources and Social Security and the Ministry of Finance on the Adjustment of Retirees' Basic Pension* in 2022, issued by the Ministry of Human Resources and Social Security (2022) No. 27, the adjustment rate of China's urban workers' basic pension will be 4% from 2021, so this paper assumes that the adjustment rate of China's urban workers' basic pension will be 4% in the period of 2021–2050.

### 4.1.11. Pension Crediting Rate

As an important part of the pension insurance system, the individual account fund has the obvious attributes of individual property rights and the characteristic of complete accumulation. In the design of the state's individual account system for pension insurance, there are clear provisions for the management of individual account funds, namely that interest must be credited to individual account funds. Prior to 2005, due to the imperfect interest rate mechanism, in most cases, the interest credited to individual accounts was determined by reference to the one-year time deposit rate, which often resulted in lower interest earnings. However, since 2005, localities have begun to gradually improve the mechanism of booked interest rates for individual pension accounts by adopting the method of calculating interest rates linked to the rate of return on investments, which has led to a significant improvement in the booked interest rates, which is usually in the range of 3% to 5% or more. From 2016 to 2020, China's Ministry of Human Resources and Social Security and Ministry of Finance jointly and uniformly issued individual account crediting rates of 8.31%, 7.12%, 8.29%, 7.61%, and 6.04%, respectively. The average crediting rate for these five years is 7.47%. Based on this, this paper sets the individual account crediting rate at 7.47% in the subsequent analysis.

### 4.1.12. Average Social Wage

The average social wage is the contribution base for urban workers' pension insurance. In the past, the contribution base for urban workers' pension insurance was the average annual salary of urban non-private sector employees in the previous year. In April 2019, the *Circular of the General Office of the State Council on the Issuance of a Comprehensive Plan for Reducing the Rate of Social Insurance Premiums* adjusted the contribution base to be the weighted average annual salary of urban private sector and urban non-private sector employees to receive the full-caliber average salary of employed persons. However, at present, China does not have a unified weighted calculation method, and each province decides according to the actual situation of the basic old-age insurance for urban workers

and employees. The average wage of urban workers in China is roughly equal to the sum of the average annual wage of employees in the urban private sector and the average annual wage of employees in the urban non-private sector, divided by 2.3. Therefore, this paper calculates the average social wage of urban workers in 2020 to be CNY 67,437.39 based on the weighted calculation of the data from the National Bureau of Statistics (NBS).

4.1.13. Government Financial Subsidies as a Proportion of Urban Workers' Basic Pension Income

Based on the data from the *2010–2020 Statistical Bulletin on the Development of Human Resources and Social Security*, we can calculate the total value of financial subsidies to the basic pension insurance fund as a proportion of the total income of the urban basic pension insurance fund for the period 2010–2020. First, we extract the data from the statistical bulletin on the amount of financial subsidies and the total income of the urban basic pension insurance fund for each year. Then, for the data in each year, the proportion of the financial subsidy to the total income of the fund is calculated. Last, these 11 weights are added together to find the average, and we can obtain an average weight of 14.09% for 2010–2020. We can assume that the proportion of financial subsidy income of China's urban workers' basic pension insurance fund to the fund's total income is guaranteed to remain unchanged from 2021 to 2050, and will still be 14.09%.

*4.2. Delayed Retirement Program Design*

In China's current pension system, there is a difference in retirement ages between men and women, which affects the balance of pension income and expenses to some extent. Specifically, male workers retire at the age of 60, while female workers retire as early as 50, and female cadres retire at 55. According to population forecasting, the male population exceeds that of females before the age of 60. However, with the aging of the population, the female population gradually surpasses the male population after the age of 60, reflecting that females generally have a longer life expectancy than males.

Given that female workers retire earlier than males and live longer, their impact on the pension fund's revenue and expenditure cannot be ignored. Therefore, considering a gradual postponement of female retirement, especially female workers, may ensure the long-term stability of the pension system.

Meanwhile, China is facing the intensification of population aging and a decrease in the labor force. With the increasing proportion of the elderly population and the corresponding decrease in the proportion of young people, the labor market will face greater pressure. Therefore, extending the retirement age overall has become a strategy that urgently needs consideration. This measure is not only helpful in alleviating the financial pressure on pensions but also provides more stable human resource support for China's economic and social development.

This paper refers to Yu and Zeng (2015)'s research on the adjustment of the legal retirement age and proposes a phased delayed retirement scheme. In the first stage, from 2022 to 2031, the retirement age of female workers will be gradually delayed from 50 to 55 years old, to align with the retirement age of female cadres; in the second stage, from 2032 to 2041, the retirement age for females will continue to be delayed, adjusted from 55 to 60 years old, equal to that of males; in the third stage, from 2042 to 2050, both male and female workers will have their retirement age delayed until 2049 when the retirement age for both genders will be 65 years old, and the retirement age will remain at 65 in 2050 [37]. This scheme aims to gradually balance the income and expenditure of the pension fund and ensure the long-term sustainability of the pension insurance system.

## 5. Calculation Results and the Analysis

*5.1. Impact of Delayed Retirement on the Income of the Pension Fund*

The above parameters are brought into the model of pension income regarding the aspects of social coordination, individual accounts, financial subsidy, and the total pension

fund income to measure the contribution of urban workers' basic pension fund, financial subsidy, and total pension fund income before and after the simulated implementation of the delayed retirement policy from 2021 to 2050, and the results of the calculations are shown in Table 1.

**Table 1.** Comparison of pension income before and after the implementation of the delayed retirement policy (CNY hundred million).

| Year | Pre-Delayed Retirement | | | Post-Delayed Retirement | | |
|---|---|---|---|---|---|---|
| | Levy Income | Financial Subsidy | Pension Income | Levy Income | Financial Subsidy | Pension Income |
| 2021 | 67,684.52 | 11,100.86 | 78,785.38 | 67,684.52 | 11,100.86 | 78,785.38 |
| 2022 | 73,098.53 | 11,988.81 | 85,087.34 | 73,098.53 | 11,988.81 | 85,087.34 |
| 2023 | 78,447.86 | 12,866.14 | 91,314.00 | 79,081.56 | 12,866.14 | 91,947.71 |
| 2024 | 84,202.00 | 13,809.87 | 98,011.88 | 85,552.08 | 13,809.87 | 99,361.95 |
| 2025 | 90,621.32 | 14,862.70 | 105,484.02 | 92,710.86 | 14,862.70 | 107,573.56 |
| 2026 | 96,839.13 | 15,882.47 | 112,721.61 | 99,708.54 | 15,882.47 | 115,591.01 |
| 2027 | 103,745.24 | 17,015.14 | 120,760.38 | 107,405.93 | 17,015.14 | 124,421.06 |
| 2028 | 111,212.13 | 18,239.77 | 129,451.90 | 115,805.36 | 18,239.77 | 134,045.13 |
| 2029 | 119,111.65 | 19,535.36 | 138,647.02 | 124,105.68 | 19,535.36 | 143,641.04 |
| 2030 | 127,624.15 | 20,931.49 | 148,555.64 | 134,584.62 | 20,931.49 | 155,516.11 |
| 2031 | 135,851.90 | 22,280.91 | 158,132.81 | 144,146.76 | 22,280.91 | 166,427.67 |
| 2032 | 144,800.08 | 23,748.49 | 168,548.57 | 153,136.63 | 23,748.49 | 176,885.12 |
| 2033 | 154,370.53 | 25,318.13 | 179,688.66 | 164,621.60 | 25,318.13 | 189,939.73 |
| 2034 | 164,712.63 | 27,014.33 | 191,726.96 | 176,920.50 | 27,014.33 | 203,934.83 |
| 2035 | 175,751.27 | 28,824.76 | 204,576.03 | 190,133.64 | 28,824.76 | 218,958.40 |
| 2036 | 185,797.92 | 30,472.50 | 216,270.42 | 202,605.26 | 30,472.50 | 233,077.76 |
| 2037 | 196,411.85 | 32,213.28 | 228,625.13 | 216,352.13 | 32,213.28 | 248,565.41 |
| 2038 | 207,211.53 | 33,984.52 | 241,196.06 | 230,829.35 | 33,984.52 | 264,813.88 |
| 2039 | 217,812.47 | 35,723.17 | 253,535.64 | 245,411.28 | 35,723.17 | 281,134.45 |
| 2040 | 228,244.31 | 37,434.09 | 265,678.39 | 260,236.66 | 37,434.09 | 297,670.74 |
| 2041 | 236,411.52 | 38,773.58 | 275,185.10 | 272,620.93 | 38,773.58 | 311,394.51 |
| 2042 | 244,924.54 | 40,169.79 | 285,094.33 | 284,887.44 | 40,169.79 | 325,057.23 |
| 2043 | 253,917.11 | 41,644.65 | 295,561.76 | 299,610.02 | 41,644.65 | 341,254.67 |
| 2044 | 263,550.18 | 43,224.56 | 306,774.74 | 315,639.76 | 43,224.56 | 358,864.32 |
| 2045 | 273,554.41 | 44,865.34 | 318,419.75 | 332,459.68 | 44,865.34 | 377,325.02 |
| 2046 | 281,322.76 | 46,139.42 | 327,462.18 | 347,021.32 | 46,139.42 | 393,160.74 |
| 2047 | 288,729.71 | 47,354.23 | 336,083.93 | 362,170.50 | 47,354.23 | 409,524.73 |
| 2048 | 296,219.03 | 48,582.54 | 344,801.57 | 377,953.57 | 48,582.54 | 426,536.11 |
| 2049 | 304,112.74 | 49,877.18 | 353,989.92 | 394,404.01 | 49,877.18 | 444,281.19 |
| 2050 | 310,013.49 | 50,844.96 | 360,858.45 | 408,099.07 | 50,844.96 | 458,944.02 |

As can be seen from Table 1, from 2021 to 2050, before and after the implementation of the delayed retirement simulation, China's urban workers' pension insurance fund's collection income, government financial subsidies, and total income all show a continuous growth trend. In particular, after the implementation of the delayed retirement policy, there is a significant increase in the growth rate of both the collected income and the total income of the fund.

Specifically, using the 2021 data as a baseline, by 2050, before the implementation of the delayed retirement policy, the levy revenue, government financial subsidy, and total fund revenue grow from CNY 6768.452 billion, CNY 1110.086 billion, and CNY 7878.538 billion to CNY 31,001.349 billion, CNY 5084.496 billion, and CNY 36,085.845 billion, respectively, with an annual average growth rate of 5.39%. After the delayed retirement policy is simulated and implemented, the same three indicators will reach CNY 40,809.907 billion, CNY 5084.496 billion, and CNY 45,894.402 billion in 2050, with average annual growth rates of 6.39%, 5.39%, and 6.26%, respectively.

It is worth noting that, after the implementation of the policy, the incremental increase in the pension fund's contribution revenues and total revenues is characterized by annual

growth, reaching CNY 9808.557 billion by 2050. This figure shows that, provided that the level of government financial subsidies remains stable, the growth of the fund's total income is largely dependent on the increase in contribution revenue.

In summary, the implementation of the delayed retirement policy has had a positive impact on increasing the income level of the basic pension insurance fund for urban workers. This result verifies our theoretical Hypothesis 1. The increase in the income level of the pension fund can effectively enhance the financial stability of the pension insurance fund.

## 5.2. Impact of Delayed Retirement on Pension Fund Expenditures

The above parameters are brought into the pension fund expenditure model to measure the expenditure of the basic pension insurance fund for urban workers before and after the simulated implementation of the delayed retirement policy from 2021 to 2050, and the results are shown in Table 2.

**Table 2.** Comparison of pension expenditures before and after the implementation of delayed retirement policy (CNY hundred million).

| Year | Pre-Delayed Retirement | Post-Delayed Retirement | Year | Pre-Delayed Retirement | Post-Delayed Retirement |
|---|---|---|---|---|---|
| 2021 | 74,483.33 | 74,483.33 | 2036 | 327,498.29 | 290,452.05 |
| 2022 | 83,155.23 | 83,155.23 | 2037 | 352,134.37 | 308,182.60 |
| 2023 | 93,721.76 | 92,356.04 | 2038 | 378,244.60 | 326,186.90 |
| 2024 | 105,533.92 | 102,624.31 | 2039 | 406,079.49 | 345,247.01 |
| 2025 | 118,289.79 | 113,786.54 | 2040 | 435,721.20 | 365,204.60 |
| 2026 | 132,239.11 | 126,003.12 | 2041 | 466,587.41 | 385,934.77 |
| 2027 | 146,024.54 | 138,068.88 | 2042 | 493,779.98 | 404,766.84 |
| 2028 | 161,089.26 | 151,106.93 | 2043 | 521,579.91 | 419,803.78 |
| 2029 | 177,893.78 | 165,474.95 | 2044 | 549,671.96 | 433,647.91 |
| 2030 | 196,054.06 | 180,927.06 | 2045 | 578,712.82 | 447,507.53 |
| 2031 | 215,453.02 | 197,324.39 | 2046 | 609,126.28 | 461,517.18 |
| 2032 | 235,219.12 | 216,999.38 | 2047 | 636,392.76 | 471,388.68 |
| 2033 | 256,459.68 | 234,055.70 | 2048 | 664,598.74 | 480,960.57 |
| 2034 | 278,772.70 | 252,092.08 | 2049 | 693,085.36 | 490,222.27 |
| 2035 | 302,271.35 | 270,838.30 | 2050 | 718,122.17 | 497,747.10 |

According to the data presented in Table 2, we can clearly observe the significant changes in the expenditures of China's basic pension fund for urban workers before and after the implementation of the delayed retirement policy. After simulating the implementation of the policy, there is a downward trend in the expenditure of the pension fund. Specifically, the effect of the policy begins to show in 2023, when the expenditure of the basic pension insurance fund for urban workers drops to CNY 9235.604 billion, a decrease of CNY 136.572 billion in comparison with the expenditure before the policy was implemented. With the gradual passage of time, this expenditure reduction shows an increasing trend year by year, and by 2050, the expenditure reduction will reach the maximum value of CNY 22,037.507 billion. This series of data fully proves that the delayed retirement policy has a significant effect on easing the pressure of the expenditure of China's basic pension insurance fund for urban workers, and can effectively curb the excessive growth of the fund's expenditure, thus verifying theoretical Hypothesis 2. The reduction in pension expenditure helps to improve the sustainability of the pension insurance fund.

## 5.3. Analysis of the Impact of Delayed Retirement on the Sustainability of Pension Funds

The above parameters are brought into the current pension fund income and expenditure gap measurement model and the cumulative balance measurement model to measure the current income and expenditure gap and cumulative balance of the pension fund before and after the simulated implementation of the delayed retirement policy from 2021 to 2050, and the results of the measurement are shown in Table 3.

**Table 3.** Income and expenditure of the basic pension insurance fund for urban workers before and after the delay in retirement (CNY hundred million yuan).

| Year | Pre-Delayed Retirement | | Post-Delayed Retirement | |
| | Difference between Current Income and Expenditure | Cumulative Balance | Current Income and Expenditure | Cumulative Balance |
|---|---|---|---|---|
| 2021 | 4302.05 | 52,619.05 | 4302.05 | 52,619.05 |
| 2022 | 1932.11 | 54,551.16 | 1932.11 | 54,551.16 |
| 2023 | −2407.76 | 52,143.40 | −408.33 | 54,142.83 |
| 2024 | −7522.04 | 44,621.36 | −3262.36 | 50,880.47 |
| 2025 | −12,805.78 | 31,815.58 | −6212.98 | 44,667.49 |
| 2026 | −19,517.50 | 12,298.08 | −10,412.11 | 34,255.38 |
| 2027 | −25,264.16 | −12,966.07 | −13,647.81 | 20,607.57 |
| 2028 | −31,637.36 | −44,603.43 | −17,061.80 | 3545.77 |
| 2029 | −39,246.76 | −83,850.19 | −21,833.91 | −18,288.15 |
| 2030 | −47,498.43 | −131,348.62 | −25,410.94 | −43,699.09 |
| 2031 | −57,320.21 | −188,668.83 | −30,896.72 | −74,595.81 |
| 2032 | −66,670.55 | −255,339.38 | −40,114.25 | −114,710.07 |
| 2033 | −76,771.02 | −332,110.40 | −44,115.97 | −158,826.04 |
| 2034 | −87,045.74 | −419,156.14 | −48,157.26 | −206,983.29 |
| 2035 | −97,695.32 | −516,851.46 | −51,879.89 | −258,863.19 |
| 2036 | −111,227.87 | −628,079.33 | −57,374.29 | −316,237.48 |
| 2037 | −123,509.23 | −751,588.57 | −59,617.18 | −375,854.66 |
| 2038 | −137,048.55 | −888,637.11 | −61,373.02 | −437,227.68 |
| 2039 | −152,543.85 | −1,041,180.96 | −64,112.56 | −501,340.24 |
| 2040 | −170,042.80 | −1,211,223.77 | −67,533.86 | −568,874.09 |
| 2041 | −191,402.31 | −1,402,626.08 | −74,540.26 | −643,414.35 |
| 2042 | −208,685.66 | −1,611,311.74 | −79,709.61 | −723,123.96 |
| 2043 | −226,018.14 | −1,837,329.88 | −78,549.11 | −801,673.07 |
| 2044 | −242,897.23 | −2,080,227.11 | −74,783.59 | −876,456.65 |
| 2045 | −260,293.07 | −2,340,520.17 | −70,182.51 | −946,639.16 |
| 2046 | −281,664.10 | −2,622,184.27 | −68,356.44 | −1,014,995.61 |
| 2047 | −300,308.82 | −2,922,493.09 | −61,863.95 | −1,076,859.56 |
| 2048 | −319,797.17 | −3,242,290.27 | −54,424.45 | −1,131,284.01 |
| 2049 | −339,095.45 | −3,581,385.72 | −45,941.08 | −1,177,225.09 |
| 2050 | −357,263.72 | −3,938,649.43 | −38,803.08 | −1,216,028.17 |

According to the data presented in Table 3, we can clearly observe that in the comparison before and after the simulated implementation of the retirement policy, the current balance of income and expenditure and the accumulated balance of the basic pension insurance fund for urban employees have changed significantly. Before the implementation of the simulation policy, by 2023, the fund's current income will not be able to meet the demand for expenditure, resulting in a funding gap of up to CNY 240.776 billion. With the worsening problem of "income not meeting expenditure", the cumulative shortfall of the fund will continue to widen and is expected to reach a staggering CNY 393,864.943 billion in 2050. However, after the implementation of the retirement policy simulation, the current shortfall in 2023 is drastically reduced to CNY 40.833 billion, which is CNY 199.943 billion less than that before the implementation of the policy. With the passage of time, the current shortfall shrinks even more significantly to CNY 31,846,064 billion in 2050. This trend clearly shows that the delayed retirement policy has reduced the financial pressure on the fund to a certain extent, and has had a positive effect on ensuring the sustainability of the pension system, validating our theoretical Hypothesis 3.

With regard to the accumulated balance, before the retirement policy simulation was introduced, a shortfall was expected to form in 2027, while after the policy simulation was introduced, the emergence of the shortfall was delayed until 2029. This change suggests that while the delayed retirement policy is able to delay the emergence of the shortfall, it does not completely eliminate the problem. In other words, the delayed retirement

policy only delays the emergence of the pension financial gap to a certain extent, rather than being the fundamental solution to the problem. Therefore, we still need to find more comprehensive and effective measures to ensure that the pension system can develop stably in the long term.

## 6. Discussion

The sustainability of pension funds has become a major concern for governments around the world in the context of aging populations and economic downturn. This paper reveals the sustainability of China's basic pension fund for urban workers by studying the role of delayed retirement reform. By building an actuarial model of pension fund income and expenditure, the sustainability of China's basic pension fund for urban workers is assessed under a series of assumptions for the period from 2021 to 2050. The results show that China's basic pension insurance for urban workers will face severe pressure in pension payments. The calculations found that if China maintains the current statutory retirement age, the pension fund will be out of balance by early 2027. This situation is a bit later than that found in previous studies [15]. This is because most of the previous studies have ignored the role of financial subsidies, and the share of total pension income accounted for by government financial subsidies has been as high as 14% on average over the past 10 years. Therefore, in this paper, by considering the social pooling account, the individual account, and the financial subsidies as a whole in the total pension income, our findings may be more in line with reality. The simulation of delayed retirement finds that, with the implementation of the delayed retirement strategy, a gap occurs in 2029, two years later than that before the delay, and significantly reduces the annual pension gap. Delaying retirement improves the solvency of the basic pension fund system for urban workers, but the pension gap does not disappear. In the long run, the retirement delay cannot fundamentally solve the pension payment crisis. The results of the general trend of the widening pension gap are consistent with the findings of Zhao et al. [38] and Tian and Zhao [6]. Based on the above empirical analysis, we suggest that the government should implement retirement system reform as soon as possible. This study assumes that some designs of retirement policy options have not yet been realized in practice. In future studies, the scenario assumptions should be adjusted according to the actual situation. Although this study focuses on China, the conclusions drawn are also valuable and provide references for other countries facing severe population aging issues and substantial pressures on pension fund expenditures [39].

Building an actuarial model for the pension fund's revenue and expenditure can help the government and relevant departments better understand the operation of the basic urban employee pension fund from 2021 to 2050, predict future trends in revenue and expenditure, and simulate the effects of delayed retirement policies. This can provide a scientific basis for policy formulation. However, actuarial models primarily focus on the financial status of the pension fund, and other aspects of the pension system, such as fairness, may require assessment through other research methods. Future research can evaluate the fairness of the system by comparing the pension treatments of different social groups and regions. Additionally, the issue of sustainable pension funds is an important global topic. Subsequent studies will select international case studies for comparative analysis to explore the best strategies for solving the sustainable pension fund issue.

## 7. Conclusions

This article employs rigorous actuarial methods to explore the impact of the delayed retirement policy on the sustainability of urban workers' pension insurance funds. This paper first theoretically analyzes how delayed retirement affects the sustainability of pension funds and proposes a series of theoretical assumptions. Then, in close conjunction with China's current basic urban workers' pension insurance system and social realities, actuarial models are constructed to simulate and predict the pension fund's financial

situation over the next three decades before and after the implementation of the delayed retirement policy.

With meticulous parameter setting, this article comprehensively calculates and comparatively analyzes the pension fund contributions, total fiscal subsidy funds, fund expenditures, annual surplus and deficit, and long-term accumulated deficits before and after the implementation of the delayed retirement policy. The research results show the following: First, the implementation of the delayed retirement policy helps to increase the pension fund contributions and total revenue. Second, delayed retirement can effectively reduce pension fund expenditures. Third, delayed retirement contributes to narrowing the annual surplus and deficit gap. Fourth, delayed retirement can postpone the appearance of the long-term accumulated deficit gap. These findings indicate that the delayed retirement policy has a significant positive effect on alleviating the payment pressure on pension funds and ensuring their sustainability [40].

However, this study also finds that although the delayed retirement policy can improve the pension fund's financial situation to some extent, it cannot completely eliminate the fund gap. Therefore, to ensure the long-term, stable, and sustainable development of the pension fund, the delayed retirement policy needs to be combined with other relevant policy measures to form a comprehensive solution.

Based on the findings of this study, we make the following policy recommendations.

Firstly, a retirement policy that combines flexibility and phased progression should be implemented. Currently, we can consider establishing a flexible retirement system, converting the legal retirement age into a flexible range, and allowing workers to choose their retirement time based on their individual circumstances within this range. When conditions are mature, we can gradually implement a phased retirement age extension plan. Based on population forecast results, China's future population structure features more males under the age of 60 and more females over the age of 60. Considering this, we can first gradually extend the retirement age for female workers from the current 50 years to 55 years, by two years each time, until it aligns with the retirement age for female cadres. Next, we can extend the retirement age for females from 55 years to 60 years, by two years each time, until it matches the retirement age for males. Finally, we can unify the retirement age to 65 years. Such a phased adjustment can not only increase the number and amount of contributors, thereby raising the income level of the pension fund but also give workers sufficient time to adapt, avoiding social shock from a sudden and drastic change.

Thirdly, a pension incentive mechanism for delayed retirement should be established. The establishment of pension incentives for delayed retirement can encourage people to delay receiving their pensions in order to reduce the pressure on pension payments. Establishing a pension incentive mechanism for delayed retirement can encourage people to defer claiming their pensions, thereby alleviating the pressure on pension payments. To achieve this, we could first increase the pension percentage. For individuals who choose to delay retirement, their annual pension percentage can be gradually increased, thus motivating them to delay retirement in order to receive a higher pension benefit. Second, we could offer a one-time bonus. A one-time retirement bonus can be provided for those who choose to delay retirement. This will increase the incentive for people to delay retirement. Third, differentiated treatments can be implemented, establishing different levels of pension incentives based on the number of years of delayed retirement and individual contributions, providing differentiated treatments. This will incentivize individuals to choose to delay their retirement for a longer period.

Fourthly, delayed retirement should be used in conjunction with other methods to ensure the continued effective operation of the pension insurance system. Within the measurement interval, the delayed retirement program can only narrow the annual pension income and expenditure gap of urban workers' pension insurance and ease the payment pressure on the pension fund in that year, but it cannot safely eliminate the gap. In order to effectively address the serious impact of population aging on pension fund expenditures, a combination of methods should be used to ensure the continued effective operation

of the urban workers' pension insurance system, such as lowering the rate of corporate contributions, reducing the burden of corporate contributions, and, at the same time, increasing the rate of compliance with corporate contributions, expanding investment channels, and increasing the rate of return on the pension insurance fund.

**Author Contributions:** Conceptualization, G.Z., Y.F. and D.Z.; methodology, G.Z. and Y.F.; software, D.Z.; validation, G.Z., Y.F. and D.Z.; formal analysis, G.Z. and Y.F.; investigation, G.Z. and D.Z.; resources, G.Z. and D.Z.; data curation, D.Z.; writing—original draft preparation, G.Z., Y.F. and D.Z.; writing—review and editing, G.Z. and D.Z.; visualization, G.Z., Y.F. and D.Z.; supervision, G.Z., Y.F. and D.Z.; project administration, G.Z. and D.Z.; funding acquisition, G.Z. and Y.F. All authors have read and agreed to the published version of the manuscript.

**Funding:** This research was funded by the National Social Science Fund of China, grant number 18BRK027, and the Basic Project of Liaoning Provincial Department of Education, grant number LJC202019.

**Institutional Review Board Statement:** Not applicable.

**Informed Consent Statement:** Not applicable.

**Data Availability Statement:** The data used in our study are from the Seventh National Population Census of China, the China Population and Employment Statistics Yearbook, the *China Statistical Yearbook*, the *Statistical Bulletin on the Development of Human Resources and Social Security*, the China Life Insurance Experience Tables, and the National Bureau of Statistics of China.

**Conflicts of Interest:** The authors declare no conflicts of interest.

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
