# Peer review of "Study on the Impact of Delayed Retirement on the Sustainability of the Basic Pension Insurance Fund for Urban Employees in China"

_sustainability, doi:10.3390/su16103969_

Round 1
Reviewer 1 Report
Comments and Suggestions for Authors
While this is an important global subject (aging population, social insurance/retirement plans), it is not appropriate in the Sustainability space. This is more relevant to governmental policy or the actuarial/insurance areas.
Comments on the Quality of English LanguageThere are some idioms that come across as being inappropriate or awkward.
Lines 36-38 need to be rewritten
Capitalize Financial Subsidies line 242
Delete "Morphing" line 253
In tables 2-4, remove "particular" and just use Year. Also, capitalize Financial in column header in table 1.
Overall, the quality of English is good.
Reviewer 2 Report
Comments and Suggestions for Authors
tis a very interesting topic, but the paper is very extensive, you have to simthetisçze it and explain in methods how did you analize data and perform the forecasting, you explain that you applied a statistic models but it have to be explained better and concise
Reviewer 3 Report
Comments and Suggestions for Authors
The authors used an actuarial model of pension insurance fund with respect to income and expenditure to simulate the impact of delayed retirement on the sustainability of basic pension insurance fund for urban employees in China, which fits well into the scope of the journal. The authors' writing represents efforts made to meet academic standards throughout the manuscript, as evidenced by the appropriate introduction, literature review, and data analysis. However, the authors may improve further the manuscript through paying attention to the use of latest literature to support or contest the research hypotheses and findings. Besides, the authors are also suggested to elaborate on the research limitations of the methods and future research plans, so as to inspire advancement in this line of research. Overall, the manuscript represents potential for publication subject to minor revisions.
Comments on the Quality of English LanguageMinor editing of English language is required to improve the accuracy of the language use and enhance the flow of the writing.
Reviewer 4 Report
Comments and Suggestions for Authors
The article offers a detailed examination of the challenges and potential solutions related to the sustainability of the Basic Pension Insurance Fund for Urban Employees in China, with a specific focus on the impact of delayed retirement policies.
Overall, the article provides valuable insights into the complex interaction between population aging, pension fund sustainability, and retirement policy reforms. It highlights the importance of considering factors such as government financial subsidies, investment channels, and compliance rates to ensure the long-term viability of pension systems.
However, there are areas where the article could be strengthened, if the authors wish to, such as providing a more in-depth theoretical analysis, offering innovative solutions to address sustainability challenges, and discussing the generalizability of the findings to other contexts.
Additionally, incorporating comparative analyses with international case studies could enrich the discussion and provide a broader perspective on pension fund sustainability strategies.
In conclusion, the article serves as a valuable contribution to the literature on pension fund sustainability and retirement policy, but there is potential for further development in terms of theoretical depth, practical recommendations, and directions for future research.
Reviewer 5 Report
Comments and Suggestions for Authors
Good and meaningful study.
Methodology is clear.
Elaboration is just right, not too much and not too less.
The manuscript is very readable.
Conclusion is well organized.
The authors may wish to note that in the pension world, the funding can be very different between a defined benefit (DB) plan and a defined contribution (DC) plan. If it is possible, remarks can be given to describe whether the majority of the pension plans in China belong to the DB type.
Comments on the Quality of English LanguageGenerally the English writing is quite good.
Occasionally there are a few sentences which are too lengthy with complex sentence structure and long phrases. These few sentences can be rewritten.
Please see below a list of typographical issues:
Line 5: The Chinese comma (頓號) should be replaced by English comma ( , ) between the superscripts 1 and 2 of the first author.
Line 18: "retirementt" was typed with double t.
Line 21: "retirementt" was typed with double t.
Line 21: "can not" should be "cannot" without space.
Line 29: You used "ageing" in the keyword but you mostly used "aging" in the main context. Both words are commonly used, but it is suggested that the use should be consistent throughout the whole manuscript. You may use search function to locate all the strings "ageing" and "aging" to check.
Line 34: Should "object" be "subject"?
Line 36-37: "China's urban workers' pension insurance fund" has too many apostrophe s. It is suggested that you can write it as "the pension insurance fund for China's urban workers".
Line 37: "appeared" should be changed to "showed", because "appear" is an intrasitive verb, i.e., it should not be followed by an object.
Line 38: "appeared" can be changed to "demonstrated".
Line 52: "argues" should be "argue".
Line 53: "ageing" vs "aging" (Both are correct, but it is better to be consistent in the whole manuscript.)
Line 63: Need to add a space after "2087".
Line 90: Delete "found that" because this phrase already appeared in Line 89.
Line 120: "ageing"
Line 137: "delayed" should be "Delayed" with capital D.
Line 142: You used "labour" here but "labor" without u in Line 143 and 144. It is better to search for the whole paper to be consistent.
Line 142: "On the one hand" should be "On one hand".
Line 156-157: "workplace" may be "workforce".
Line 182: "of the growth rate" should be "has a growth rate"?
Line 185: "so on" should be "so the following result".
Equation (3): Delete "\times (1 + \delta)^{i-b}".
Line 186: "Where" should be "where" with small w.
Line 186: "in the year t" should be "in year t".
Line 187: "in the year t-1" should be "in year t-1".
Line 188: "in the year t-i+b" should be "in year t-i+b".
Line 190: "in the year t" should be "in year t".
Line 193: "in the year t" should be "in year t".
Line 194: "of the year t" should be "of year t".
Line 194: "of the year t-1" should be "of year t-1".
Line 195: "of the year t" should be "of year t".
Line 196: "Where" should be "where" with small w.
Line 197: "in the year t" should be "in year t".
Line 197-198: "of the year t" should be "of year t".
Line 198: "of the year t" should be "of year t".
Line 200: "in the year t" should be "in year t".
Line 206: "Where" should be "where" with small w.
Line 206: The colon ( : ) after "where" should be deleted.
Line 207: "in the year t" should be "in year t".
Line 209: "in the year t" should be "in year t".
Line 211: "in the year t" should be "in year t".
Line 213: Add "in year t" in the end.
Line 218: Add "subscript" before m and delete "in the lower corner".
Line 218: Add "subscript" before f.
Line 222: "in the year t" should be "in year t".
Line 228: "Where" should be "where" with small w.
Line 228: The colon ( : ) after "where" should be deleted.
Line 228: "in the year t" should be "in year t".
Line 231: "in the year t" should be "in year t".
Line 232: "in the year t" should be "in year t".
Line 235: Add "in year t" in the end.
Equation (9): Delete "\times (1 + \delta)^{i-b}".
Equation (10): Delete "\times (1 + \delta)^{i-b}" twice.
Equation (10): In the upper limit of the first summation, the subscript m should be italic.
Line 240: Add "subscript" before m and delete "in the lower corner".
Line 240: Add "subscript" before f.
Line 249: "in the year t" should be "in year t".
Line 250: Add "the proportion of" after "Then".
Line 251: "in the year t" should be "in year t".
Line 253: "in the year t" should be "in year t".
Equation (13): The t in G(t) should be italic.
Line 259: "in the year t" should be "in year t".
Line 260: "of the year t" should be "of year t".
Line 261: "of the year t-1" should be "of year t-1".
Line 261: "of the year t" should be "of year t".
Line 263: "Where" should be "where" with small w.
Line 267: "Where" should be "where" with small w.
Line 267: Should omega just be w?
Equation (16): There is no need to have parentheses for i-M in the index.
Line 271: "Where" should be "where" with small w.
Line 275-276: It is not too reader friendly to say "in the year before the year in which he or she retired". Do you mean "just before the retirement year"?
Line 276: R needs to be italic as in math mode.
Equation (17): The p in the numerator should be capital P?
Line 277: "Where" should be "where" with small w.
Equation (19): In the lower limit of the first summation, the subscript m should be italic.
Line 287-293: These sentences are a bit clumsy to read due to the long phrases with many similar words. Please check whether there is a more tidy way to describe the situation. Currently, it is one long sentence. Perhaps you can break that into two different sentences.
Line 294: "Where" should be "where" with small w.
Line 302: The t should be italic.
Line 303: The first "and" can be replaced by a comma.
Note: "A and B and C" should be written as "A, B and C".
Line 306: Add a comma after 2050.
Line 308-309: "Firstly, the number of the starting population." is not a complete sentence, as you can see there is no main verb here. Perhaps change this to a single line as sub-heading like "Step 1: Set the number of the starting population".
Line 313: Similarly, perhaps change "Secondly, Setting of Fertility Patterns." to a single line as sub-heading like "Step 1: Set the fertility patterns".
Line 315: Add "from" after "age group"?
Line 316: Perhaps change "Thirdly, setting of fertility level." to a single line as sub-heading like "Step 3: Set the fertility level".
Line 319: "of childbearing age women" should be "of women in their childbearing age".
Line 319: "wishes" should be "wish".
Line 321: "women of childbearing age" should be "women in childbearing age".
Comparing Lines 317, 319, 322 and 323, why sometimes it is quoted as "1.8" while sometimes it is "1.80"?
Line 323: Perhaps change "Fourthly, Setting of Birth Sex Ratio." to a single line as sub-heading like "Step 4: Set the birth sex ratio".
Line 330: Add a space after (2021).
Line 331: Perhaps change "Fifthly, the migration pattern and the migration level." to a single line as sub-heading like "Step 5: Set the migration pattern and the migration level".
Line 335: Perhaps change "Sixthly, life expectancy." to a single line as sub-heading like "Step 6: Set the life expectancy".
Figure 1: The graph legend of "a male" and "females" are not consistent.
Line 345: There should be no period (full stop).
Line 347: Add "the" before "data".
Line 352: Is there something missing between "for males is" and "The retirement age for men is"??? Do you mean male workers or cadres?
Line 359: Need a space after "Assuming".
Line 361: "Where" should be "where" with small w.
Line 383: "will" should be "would", because the years 2010-2020 have passed.
Line 384: Add a comma before "respectively".
Line 387: The tilde ( ~ ) should be replaced by the word "to".
Line 394: Add a space after (2017).
Line 395: "rate growth" should be "growth rate".
Line 402: Why there is capitalization of P and Y in "Past Years"?
Line 404: Add a comma before "respectively".
Line 414: Need an extra space before AND after "et al.".
Line 414: "will" should be "would".
Line 415: "will" should be "would".
Line 479: "urban private sector employees" should be "employees in the urban private sector".
Line 480: "urban non-private sector employees" should be "employees in the urban non-private sector".
Line 489: "for each year's data" should be "for the data in each year"
Figure 2: The legend should include the yellow curve and the orange curve.
Line 507: Delete the space after 2021.
Line 509: "Expenditures" should be "Expenditure".
Table 1: "particular year" should be "Year".
Table 1: "financial subsidy" should be "Financial subsidy".
Table 1: The levy income from 2027 to 2050, should the decimal point be removed?
Line 511: "From the data in Table 1" should be just "In Table 1".
The values are projected values based on your model, so maybe it is better not called them "data" which would feel more like historical real data.
Line 516: "the data in Table 1" should be just "Table 1".
Line 532: "the data in Table 1" should be just "Table 1".
Line 551: "is not the same" should be "are not the same".
Line 556: "graphic" should be "graph".
Line 557: "left" should be "on the left".
Line 557: "right" should be "on the right".
Line 557: "upper old" should be "elderly at the top".
Line 557: "lower young" should be "youngster at the bottom".
Lines 568, 572, 573, 575: Sometimes you used "12th", "Fourteenth", "23rd", "14th". Maybe it is better to check the whole article to be consistent.
Line 579: "retirementt" should be "retirement".
Line 580-581: I am not very sure I understand the meaning of "two-year delay of one year".
Figure 3: It is better to change the Chinese words "年份:" to English "Year", so as other Chinese words in the diagram, such as 男性, 女性, 千万.
Line 587: Add a space after (b).
Line 592: "retirementt" should be "retirement".
Line 593: "retirementt" should be "retirement".
Table 2: The two columns of "particular year" should be "Year".
Line 605: "retirementtt" should be "retirement".
Line 605: The full stop (period) should be a comma.
Line 605: "After" should be "after".
Line 610: "was" should be "will be".
Line 611: "was" should be "will be".
Line 622: "retirementtt" should be "retirement".
Line 623: "retirementtt" should be "retirement".
Table 3: The two columns of "particular year" should be "Year".
Line 629: "retirementtt" should be "retirement".
Line 641: It is an incomplete sentence, not sure what you would like to say.
Line 645: "and gradually realizing" should be "would be achieved and"??? The sentence is a bit complicated. Maybe it should be rewritten clearly.
Table 4: "particular year" should be "Year".
Table 4: "Cumulative balance" for the pre-delayed retirement should be moved down for better alignment.
Line 669: "retirementtt" should be "retirement".
Line 672: "can't" is better written in a formal form of "cannot".
Line 677: Delete "an".
Line 685: "later than previous studies found" should be "later than that in the previous studies found".
Line 691: "retirementtt" should be "retirement".
Line 692: "than before" should be "than that before".
Line 696: "et al" should be "et al.".
Line 712: "is measured" should be "are measured"?
Line 713: "retirementtt" should be "retirement".
Line 716: "retirementtt" should be "retirement".
Line 718: What is "fund's sources of funds"? Is it just "sources of funds"?
Line 722: "doesn't" is better written in the formal form as "does not".
Line 722: "measurement interval" may be better described as "measurement time interval"?
Line 728: "ageing" vs "aging".
Line 731: "not only affects" should be "it does not only affect".
Line 731: "will also" should be "but it will also".
Line 735: "interval" is "time interval"?
Line 765: "retirementtt" should be "retirement".
Line 772: There is a strange large spacing before "expanding".
Line 774: Need a space before the first "F.Y.".
Line 787: The capitalization of S and B in "Statistical bulletin" may need some attention to check.
Line 792: Need a space before 19.
Line 794: Need a space before 11.
Line 800: For reference list, should "et al." be used or the full list of authors should be shown?
Line 801: "(5)" seems strange?
Line 810: For reference list, should "et al." be used or the full list of authors should be shown?
Line 819: Please check the capitalization of a, s and p in "Journal of aging & social policy".
Line 822: Need a space before 11.
Line 828: Need a space before 29.
Line 831: Need a space before 1.
Line 852: The colon ( : ) after 10 is not consistent with other references.
Line 856: Need a space before the first "Y".
Line 857: (04) is strange.
Line 860: For reference list, should "et al." be used or the full list of authors should be shown?
Line 860: "bayesian" should be "Bayesian" with capital B.
Line 863: For reference list, should "et al." be used or the full list of authors should be shown?
Line 864: The colon ( : ) is strange.
Line 871: For reference list, should "et al." be used or the full list of authors should be shown?
Line 875: (06) is strange.
Line 878: Need the full name of "J. Macroecon.".
Round 2
Reviewer 1 Report
Comments and Suggestions for Authors
Although, I still believe that a policy journal would be a better placement, you are correct that Sustainability does cover human social sustainability. Thank you for following the other suggestions.
Reviewer 2 Report
Comments and Suggestions for Authors
text is now slightly clear, but I think tha you have to be more concise, reduce sections 3, 4 and 5, possoible include the complete explanations in supplementary material
Round 3
Reviewer 2 Report
Comments and Suggestions for Authors
now text appear better, I suggest delete figures 1, 2, 3 due they are not neccesary to understand text, and place tables 1, 2 3 in supplementary material
